# Efficient Multi-agent Communication via Self-supervised Information Aggregation

**Cong Guan**[1,*] **Feng Chen**[1,*] **Lei Yuan**[1,2], **Chenghe Wang**[1],
**Hao Yin**[1], **Zongzhang Zhang**[1], **Yang Yu**[1,2,†]
[1] National Key Laboratory for Novel Software Technology, Nanjing University
[2] Polixir Technologies
{guanc, chenf, yuanl, wangch, yinh}@lamda.nju.edu.cn
{zzzhang, yuy}@nju.edu.cn

## Abstract

Utilizing messages from teammates can improve coordination in cooperative Multi-agent Reinforcement Learning (MARL). To obtain meaningful information for decision-making, previous works typically combine raw messages generated by teammates with local information as inputs for policy. However, neglecting the aggregation of multiple messages poses great inefficiency for policy learning. Motivated by recent advances in representation learning, we argue that efficient message aggregation is essential for good coordination in MARL. In this paper, we propose **M**ulti-**A**gent communication via **S**elf-supervised **I**nformation **A**ggregation (MASIA), with which agents can aggregate the received messages into compact representations with high relevance to augment the local policy. Specifically, we design a permutation invariant message encoder to generate common information aggregated representation from raw messages and optimize it via reconstructing and shooting future information in a self-supervised manner. Each agent would utilize the most relevant parts of the aggregated representation for decision-making by a novel message extraction mechanism. Empirical results demonstrate that our method significantly outperforms strong baselines on multiple cooperative MARL tasks for various task settings.

## 1 Introduction

Multi-Agent Reinforcement Learning (MARL) [17] has attracted widespread attention [17, 8] recently, achieving remarkable success in many complex domains [1], such as traffic signal control [9], autonomous driving [59], and droplet control [24]. For better coordination on further applications, some issues like non-stationarity [33], scalability [3] remain to be solved. To solve the non-stationarity caused by the concurrent learning of multiple policies and scalability as the agent number increases, most recent works on MARL adopt the *Centralized Training and Decentralized Execution* (CTDE) [20, 27] paradigm, which includes both value-based methods [42, 36, 46, 53] and policy gradient methods [14, 26, 49, 52]. Under the CTDE paradigm, however, the coordination ability of the learned policies can be fragile due to the partial observability in the multi-agent environment, which is a common challenge in many multi-agent tasks [29]. While recurrent neural networks could in principle relieve this issue by conditioning the policy on action-observation history [16], the uncertainty of other agents (e.g., states and actions) at execution time can result in catastrophic miscoordination and even sub-optimality [48, 6].

---

*Equal Contribution
†Corresponding Author

36th Conference on Neural Information Processing Systems (NeurIPS 2022).

Communication shows great potential in solving these problems [55, 60], with which agents can share information such as observations, intentions, or experiences to stabilize the learning process, leading to a better understanding of the environment (or the other agents) and better coordination as a result. Previous communication methods either focus on generating meaningful information [48, 19, 57] for the message senders, or design techniques such as attention mechanism [4, 31], message gate [28, 6] to filter the most relevant information on raw received messages. These approaches treat the received information as a black box and tacitly assume that policy networks can automatically extract the most critical information from multiple raw messages during policy learning. On this occasion, with the only signal given by reinforcement learning, the extraction process may be reasonably inefficient, especially in complex scenarios.

Motivated by recent advances in state representation learning [38, 22], which reveals that auxiliary representation objectives could facilitate policy learning [12], we aim at efficiently aggregating information as compact representations for policy by designing a novel communication framework **M**ulti-**A**gent communication via **S**elf-supervised **I**nformation **A**ggregation (MASIA). Specifically, representations are optimized through self-supervised objectives, which encourages the representations to be both abstract of the true states and predictive of the future information. Since agents are guided towards higher cumulative rewards during policy learning, correlating representations with true states and future information could intensify the learning signals in policy learning. In this way, the efficiency of policy learning could be encouraged. Also, considering that permutation invariance of representations can also promote efficiency, we design a self-attention mechanism to maintain the invariance of obtained representations. We also design a network that weighs the aggregated representation for individual agents to derive unique and highly relevant representation to augment local policies. To evaluate our method, we conduct extensive experiments on various cooperative multi-agent benchmarks, including Hallway [48], Level-Based Foraging [34], Traffic Junction [4], and two maps from StarCraft Multi-Agent Challenge (SMAC) [48]. The experimental results show that MASIA outperforms previous approaches, strong baselines, and ablations of our method.

Our main contributions are:

- We propose a novel framework that uses a message aggregation network to extract from multiple messages generated by various teammates, with which we acquire a permutation invariant information aggregation representation. Agents can then use a novel *focusing network* to extract the most relevant information for decision-making.
- Two representation objectives are introduced to make the information representation *compact* and *sufficient*, including the state reconstruction and multi-step future states prediction.
- Sufficient experimental results on various benchmarks and communication conditions demonstrate that our proposed approach significantly improves the communication performance, and visualization results further reveal why it works.

## 2   Problem Formulation

This paper considers a fully cooperative MARL communication problem, which can be modeled as Decentralised Partially Observable Markov Decision Process under Communication (Dec-POMDP-Com) [32] and formulated as a tuple $\langle \mathcal{N}, \mathcal{S}, \mathcal{A}, P, \Omega, O, R, \gamma, \mathcal{M} \rangle$, where $\mathcal{N} = \{1, \ldots, n\}$ is the set of agents, $\mathcal{S}$ is the set of global states, $\mathcal{A}$ is the set of actions, $\Omega$ is the set of observations, $O$ is the observation function, $R$ represents the reward function, $\gamma \in [0, 1)$ stands for the discounted factor, and $\mathcal{M}$ indicates the set of message. At each time step, due to partial observability, each agent $i \in \mathcal{N}$ can only acquire the observation $o_i \in \Omega$ drawn from the observation function $O(s, i)$ with $s \in \mathcal{S}$, each agent holds an individual policy $\pi(a_i \mid \tau_i, m_i)$, where $\tau_i$ represents the history $(o_i^1, a_i^1, \ldots, o_i^{t-1}, a_i^{t-1}, o_i^t)$ of agent $i$ at current timestep $t$, and $m_i \in \mathcal{M}$ is the message received by the agent $i$. The joint action $\boldsymbol{a} = \langle a_1, \ldots, a_n \rangle$ leads to next state $s' \sim P(s' \mid s, \boldsymbol{a})$ and the global reward $R(s, \boldsymbol{a})$. The formal objective is to find a joint policy $\boldsymbol{\pi}(\boldsymbol{\tau}, \boldsymbol{a})$ to maximize the global value function $Q_{\text{tot}}^{\boldsymbol{\pi}}(\boldsymbol{\tau}, \boldsymbol{a}) = \mathbb{E}_{s, \boldsymbol{a}} \left[ \sum_{t=0}^{\infty} \gamma^t R(s, \boldsymbol{a}) \mid s_0 = s, \boldsymbol{a_0} = \boldsymbol{a}, \boldsymbol{\pi} \right]$, with $\boldsymbol{\tau} = \langle \tau_1, \ldots, \tau_n \rangle$. As each agent can behave as a message sender as well as a message receiver, this paper considers learning useful message representation in the received end, and agents only use local information $o_i$ as message to share within the team.

We optimize the policy by value-based MARL, where deep Q-learning [30] implements the action-value function $Q(s, \boldsymbol{a})$ with a deep neural network $Q(\boldsymbol{\tau}, \boldsymbol{a}; \boldsymbol{\theta})$ parameterized by $\boldsymbol{\theta}$. This paper follows

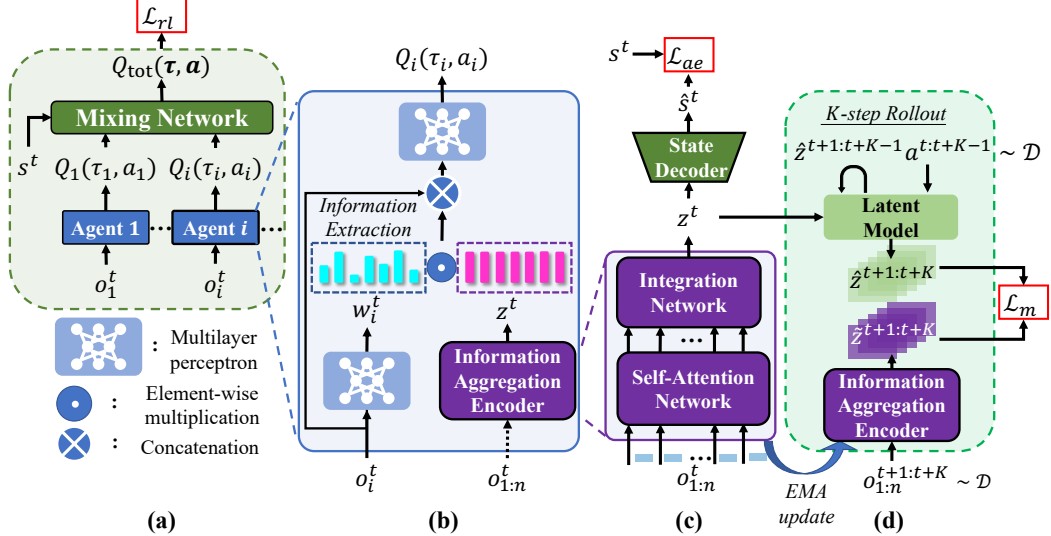

Figure 1: Structure of MASIA. (a) The overall architecture. (b) Information aggregation and extraction. (c) Information aggregation optimization. (d) Transition model learning.

the CTDE paradigm. In the centralized training phase, deep Q-learning uses a replay memory $\mathcal{D}$ to store the transition tuple $\langle \boldsymbol{\tau}, \boldsymbol{a}, r, \boldsymbol{\tau}' \rangle$. We use $Q(\boldsymbol{\tau}, \boldsymbol{a}; \boldsymbol{\theta})$ to approximate $Q(s, \boldsymbol{a}; \boldsymbol{\theta})$ to relieve the partial observability. Thus, the parameters $\boldsymbol{\theta}$ are learnt by minimizing the expected Temporal Difference (TD) error:

$$\mathcal{L}(\boldsymbol{\theta}) = \mathbb{E}_{(\boldsymbol{\tau}, \boldsymbol{a}, r, \boldsymbol{\tau}') \in \mathcal{D}} \left[ \left( r + \gamma V \left( \boldsymbol{\tau}'; \boldsymbol{\theta}^- \right) - Q(\boldsymbol{\tau}, \boldsymbol{a}; \boldsymbol{\theta}) \right)^2 \right],$$

where $V \left( \tau'; \boldsymbol{\theta}^- \right) = \max_{\boldsymbol{a}'} Q \left( \boldsymbol{\tau}', \boldsymbol{a}'; \boldsymbol{\theta}^- \right)$ is the expected future return of the TD target and $\boldsymbol{\theta}^-$ are parameters of the target network, which will be periodically updated with $\boldsymbol{\theta}$.

## 3 Method

In this paper, we propose efficient Multi-Agent communication via Self-supervised Information Aggregation (MASIA), a novel multi-agent communication mechanism for promoting cooperation performance. Redundant communications could increase the burden of information processing for each agent to make decisions and pose new challenges for information extraction since plenty of irrelevant information is contained in raw messages. To design an efficient communication mechanism, we believe two properties are of vital importance - *sufficiency* and *compactness*, where sufficiency means a rich amount of information, and compactness calls for higher information density.

To meet the standard of sufficiency, a global encoder, which we call Information Aggregation Encoder (IAE), is shared among agents to aggregate the information broadcasted by agents into a common representation. With proper training, this representation could reflect the global observation so that each agent could obtain sufficient information from it to make decisions. As for compactness, we first design an auxiliary loss on the global representation to correlate it with the policy learning process, and make each agent only focus on the part of the representation related to its performance and coordination by the designed focusing network through excluding the unrelated parts. The entire framework of our method is shown in Fig. 1. We introduce the aggregation and extraction process in Sec. 3.1, and the objectives for compactness and sufficiency in Sec. 3.2. Also, a description of the training and execution process flow can be found in Appendix A.6.

### 3.1 Information Aggregation and Extraction

**Information Aggregation.** Believing that the true state should be reflected from combined messages, we design the aggregation encoder to be capable of subsuming all the messages sent from agents. Also, the communication system in multi-agent systems is flexible and permutation invariant in nature,

which calls for a permutation invariant structure for the aggregation encoder. Based on these beliefs, we apply a self-attention mechanism to aggregate multiple messages from different teammates:

$$\boldsymbol{Q}, \boldsymbol{K}, \boldsymbol{V} = \texttt{MLP}_{Q,K,V}([o_1^t, \ldots, o_i^t, \ldots, o_n^t]), \tag{1}$$

$$\boldsymbol{H} = \texttt{softmax}(\frac{\boldsymbol{Q}\boldsymbol{K}^T}{\sqrt{d_k}})\boldsymbol{V}, \tag{2}$$

where the learnable matrices $Q$, $K$, and $V$ transform the perception from all agents into the corresponding query $\boldsymbol{Q}$, key $\boldsymbol{K}$, and value $\boldsymbol{V}$, which are the concepts defined in the attention mechanism [44]. Specifically, each row vector of $\boldsymbol{H}$ can be seen as a querying result of one agent for all available information, and the hidden state $\boldsymbol{H}$ will be fed into the subsequent integration network to finally obtain the output aggregated representation $z^t$. A detailed discussion about the design of the integration network can be found in Appendix A.3.

In the centralized training phase, we use the aggregated representation $z^t$ as extra inputs in addition to the individual observation to feed the value function. Since $z^t$ contains the information required to determine the true state, taking $z^t$ as extra inputs could reduce the uncertainty about the environment states and produce better estimations on the Q-value for value functions under any value-based policy learning algorithm.

**Information Extraction.** Similar to the decision process of human beings, global messages are usually redundant for an individual agent to make good coordination in communication systems. For example, on the task of Traffic Junction [4], one natural idea is that the information of neighboring cars are more important for agents to perceive than those distant ones, and the unrelated information in global message may sometimes even confuse the agents and impede the learning when the map is large. A toy experiment shown in Fig. 2 supports this idea. We apply the QMIX algorithm in Traffic Junction tasks with different

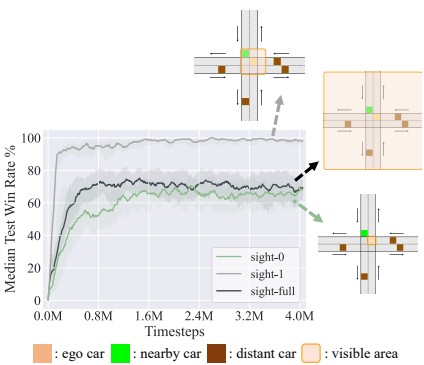

Figure 2: A toy experiment for information redundancy on the task of Traffic Junction.

sight settings. The results show that the agents learn better policies when in a small-sight setting (sight-1) than both in a super-limited-sight setting (sight-0) and full-sight setting (sight-full).

To make each agent capable of deciding its own perceptive area, we employ the **focusing network** to weigh the aggregated representations for each agent. The focusing network is designed as a Multi-Layer Perception (MLP) with the Sigmoid output activation function to ensure that each dimension of $w_i^t$ is bounded between 0 and 1. By taking element-wise multiplication with $z^t$, a unique representation could be distilled for individual agents. In this way, if the focusing network produces higher weights on specific dimensions, changes in aggregated representations on these dimensions would be more significant and thus, the agent would be more sensitive to aggregated representation on these parts. On the contrary, if some near-zero weights are outputted on some dimensions, information on those dimensions would be filtered out. In particular, although the information extraction process can, to some extent, reflect the specificity of each agent, we stress the local information by feeding it into the subsequent network together with the extracted representation.

## 3.2 Information Aggregation Representation Optimization

As for the learning process of aggregated representation, we consider two typical objectives in global encoder training: reconstruction and multi-step prediction, which constrain the representations produced by the global encoder to be sufficient and compact, respectively. For the reconstruction objective, we employ an additional decoder, which aims to reconstruct the global state by the aggregated representation to allow self-supervision on the global encoder. Specifically, the decoder is optimized together with the aggregation encoder by reconstructing global states $s^t$ from messages $\boldsymbol{o}^t$:

$$\mathcal{L}_{ae}(\theta, \eta) = \mathbb{E}_{\boldsymbol{o}^t, s^t} \|g_\eta(z^t) - s^t\|_2^2, \quad z^t = f_\theta(\boldsymbol{o}^t), \tag{3}$$

where $f_\theta, g_\eta$ denote the encoder network parameterized by $\theta$ and the decoder network parameterized by $\eta$, respectively. This loss term resembles a classical auto-encoder loss, while the decoder here is

not to reconstruct the input, but to recover the global state from representations instead. By utilizing this loss, we guide the encoder to extract observational features that can help infer the global state and let $z^t = f_\theta(\boldsymbol{o}^t)$ be a sufficient representation.

As for the multi-step prediction objective, we constrain the produced representation to be predictive of future information. Specifically, we design a transition model $h_\psi : \mathcal{Z} \times \mathcal{A}^n \to \mathcal{Z}$ parameterized by $\psi$ as auxiliary model, which predicts the aggregated representation $z^{t+1}$ on next step $t+1$ through the aggregated representation $z^t$ and joint action $\boldsymbol{a}^t$ on current step $t$. We regress the predicted aggregated representation after $k$-step rollout on the actual aggregated representation of future messages $\boldsymbol{o}^{t+k}$, updating both the aggregation encoder and the auxiliary model via the multi-step prediction loss:

$$\mathcal{L}_m(\theta, \psi) = \mathbb{E}_{\boldsymbol{o}^t, s^t, \boldsymbol{a}^t, \ldots, \boldsymbol{a}^{t+K-1}, \boldsymbol{o}^{t+K}, s^{t+K}} \left[ \sum_{k=1}^{K} \| \hat{z}^{t+k} - \tilde{z}^{t+k} \|_2^2 \right], \tag{4}$$

$$\hat{z}^{t+1} = h_\psi(\tilde{z}^t, \boldsymbol{a}^t), \tag{5}$$

$$\hat{z}^{t+k} = h_\psi(\hat{z}^{t+k-1}, \boldsymbol{a}^{t+k-1}), \quad k = 2, \ldots, K, \tag{6}$$

$$\tilde{z}^{t+k} = f_\theta(\boldsymbol{o}^{t+k}), \quad k = 0, \ldots, K. \tag{7}$$

To further stabilize the learning process, we apply the double network technique, which employs two networks with the same architecture but different update frequencies, for the aggeration encoder. The target network is updated via Exponential Moving Average (EMA) like in SPR [38]. By forcing the aggregated representation to be predictive of its future states, the aggregated representation could be more correlated with the information required for its decision-making, which meets the compactness standard. Combining these two objectives allows the aggregation encoder to extract more helpful information for agents to coordinate better. To improve the capability of information extraction on individual agents, we also enhance the learning process of these components with an RL objective. Specifically, we consider minimizing the TD loss:

$$\mathcal{L}_{rl}(\theta, \phi) = \mathbb{E}_{(\boldsymbol{\tau}, \boldsymbol{a}, r, \boldsymbol{\tau}') \in \mathcal{D}} \left[ \left( r + \gamma \max_{\boldsymbol{a}'} Q_{\text{tot}} \left( \boldsymbol{\tau}', \boldsymbol{a}'; \theta^-, \phi^- \right) - Q_{\text{tot}}(\boldsymbol{\tau}, \boldsymbol{a}; \theta, \phi) \right)^2 \right], \tag{8}$$

where $Q_{\text{tot}}$ is computed with individual $Q$-values. The computation of Q-values is actually dependent on the specific value-based learning algorithm. We apply it to prevalent methods, including VDN [42], QMIX [36], and QPLEX [46]. Moreover, the updating of the focusing network is coupled to the RL objective, making the weights produced by the focusing network could be task-sensitive, which could also facilitate policy learning.

## 4 Experiment

We conduct experiments on various benchmarks with different communication request levels[3]. Specifically, we aim to answer the following questions in this section: 1) How does our method perform when compared with multiple baselines in various scenarios (Sec. 4.1)? 2) What kind of knowledge has been learned by the information aggregation encoder (Sec. 4.2)? 3) How can the information extraction module extract the most relevant information for the individual from the learned embedding space (Sec. 4.3)? 4) Can MASIA be applied to different value decomposition baselines to improve their coordination ability and robustness in various communication conditions (Sec. 4.4)? We compare MASIA against a variety of baselines, including communication-free methods and some state-of-the-art communication approaches. QMIX [36] is a strong communication-free baseline, and we use the implementation by PyMARL[4] for comparison, which has shown excellent performance on diverse multi-agent benchmarks [37]. TarMAC utilizes an attention mechanism to select messages according to their relative importance. The implementation we used is provided by [48], denoted as TarMAC + QMIX. NDQ [48] aims at learning nearly decomposable Q functions via generating meaningful messages and communication minimization. TMC [57] applies a temporal smoothing technique at the message sender end to drastically reduce the amount of information exchanged between agents. For the ablation study, we design a baseline only different in the communication protocol, which adopts a full communication paradigm, where each agent gets message from all other teammates at each timestep, denoted as FullComm.

---

[3]The codes are available at https://github.com/chenf-ai/MASIA

[4]Our experiments are all based on the PyMARL framework, which uses SC2.4.6.2.6923.

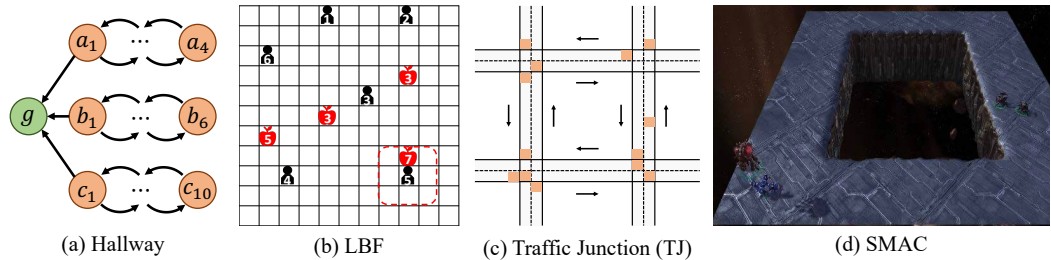

(a) Hallway      (b) LBF      (c) Traffic Junction (TJ)      (d) SMAC

Figure 3: Multiple benchmarks used in our experiments.

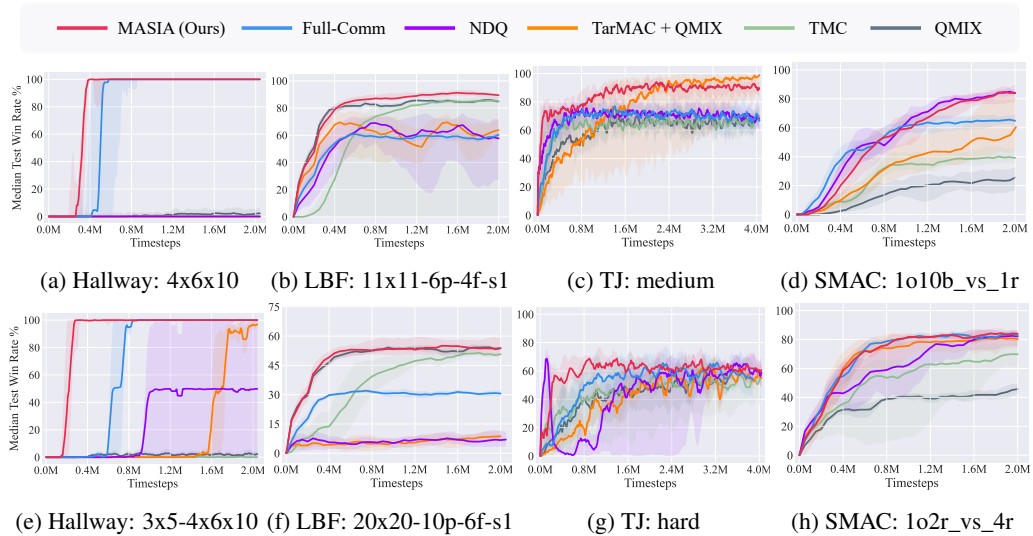

(a) Hallway: 4x6x10    (b) LBF: 11x11-6p-4f-s1    (c) TJ: medium    (d) SMAC: 1o10b_vs_1r

(e) Hallway: 3x5-4x6x10    (f) LBF: 20x20-10p-6f-s1    (g) TJ: hard    (h) SMAC: 1o2r_vs_4r

Figure 4: Performance comparison with baselines on multiple benchmarks.

We evaluate our proposed method on multiple benchmarks shown in Fig. 3. Hallway [48] is a cooperative environment under partial observability, where $m$ agents are randomly initialized at different positions and required to arrive at the goal $g$ simultaneously. We consider two scenarios with various agents and different groups, and different groups have to arrive at different times. Level Based Foraging (LBF) [34] is another cooperative partially observable grid world game, where agents coordinate to collect food concurrently. Traffic Junction (TJ) [4] is a popular benchmark used to test communication ability, where many cars move along two-way roads with one or more road junctions following the predefined routes, and we test on the medium and hard maps. Two maps named 1o2r_vs_4r and 1o10b_vs_1r from SMAC [48] require the agents to cooperate and communicate to get the position of the enemies. For evaluation, all results are reported with median performance with $95\%$ confidence interval on 5 random seeds. Details about benchmarks, network architecture and hyper-parameter choices of our method are all presented in Appendices A.1, and A.3, respectively.

## 4.1 Communication Performance

We first compare MASIA against multiple baselines to investigate the communication efficiency on various benchmarks. As illustrated in Fig. 4, MASIA achieves the best performance with low variance on all benchmarks, indicating MASIA's strong applicability in scenarios with various difficulties. In Hallway (Fig. 4a & Fig. 4e), where frequent communications are required for good performance (method without communication such as QMIX fails), other communication methods such as TarMAC, NDQ, and TMC achieve low performance or even fail in this environment. This indicates that inappropriate message generation or message selection would injure the learning process. We believe the reason why FullComm succeeds is that there is hardly any redundancy in agents' observations in Hallway. Our MASIA also succeeds in this environment, showing superiority over others. The dominating performance of MASIA is even more significant in extended Hallway

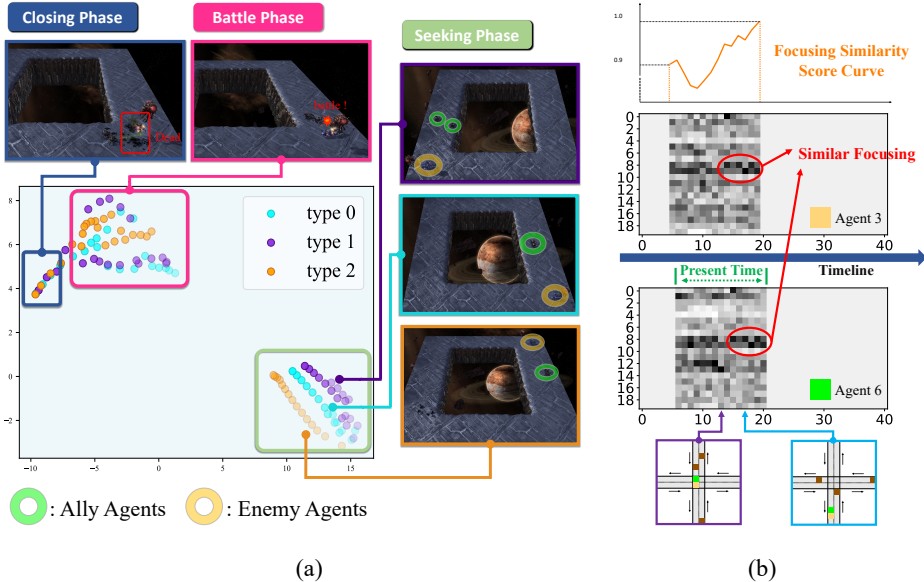

(a)                          (b)

Figure 5: (a) Information aggregation visualization. Each plotted dot represents an aggregated representation. The three different colors respectively represent three different initialization situations for the enemy entities. For example, type 0 shows the case when the enemies initialize at the lower right corner. To distinguish aggregated representations on different timesteps, we mark larger timesteps by darker shades of the dots. (b) Visualization of variations of selected agents' focus weight $w_i^t$ in a single episode. We use the horizontal axes for timesteps in one single episode and vertical axes for dimensions of the aggregated representation. The weight is reflected through luminance, and the darker the cell, the larger the weight.

(Fig. 4e), where agents are separated into different groups. In this environment, MASIA can help agents extract information about their teammates and learn a coordination pattern more efficiently. In LBF (Fig. 4b & Fig. 4f), existing communication-based MARL methods like NDQ, Fullcomm, and TarMAC struggle due to the sparsity of rewards, especially when the foods are more sparsely distributed (Fig. 4f). In contrast to the performance of QMIX in Hallway, QMIX performs well in LBF, which is attributed to the fact that agents can observe the grids near them, and the mixing network of QMIX can help improve the coordination ability of the fixed group of agents in the training phase. Our method achieves comparable performance with QMIX and TarMAC, showing its strong coordination ability even in sparse reward scenarios. In Traffic Junction (Fig. 4c & Fig. 4g), TarMAC and NDQ have high variance due to the instability of their messages, while MASIA gains high sample efficiency and can generate steady messages since it aims to reconstruct the state. On the SMAC benchmarks (Fig. 4d & Fig. 4h), we test on two complex scenarios requiring communication to succeed, where one overseer is in active service to get the information of the enemies. Messages are demanded since the agents have limited sight, so other teammates need the overseer's messages to identify the enemies' positions. Our method MASIA can maintain the high efficiency of learning and always have competitive performance when converged, which is superior to other baselines.

## 4.2 Insights into Information Aggregation Encoder

To determine what kind of knowledge the encoder has learned through training, we conduct a visualization analysis on the map 1o2r_vs_4r from SMAC to demonstrate the information contained in the aggregation representation $z^t$. We project the aggregation representation vectors into two-dimensional plane by t-SNE [43] in Fig 5a. We take trajectories from 3 scenarios of different types of initialization under a task where agents have to seek the enemy at the start, discover the enemy, and finally battle with it for better performance. It can be observed that (1) the aggregated representations could be well distinguished by phases. Projected representations in the seeking phase are far from those in the battle phase and closing phase. This implies that our learned representations could well reflect the true states. (2) the aggregated representations are first divergent in the seeking phase

when enemies have been initialized, but become increasingly interlaced later until the closing phase, when enemies have been wiped out after a fierce battle. Since enemies are highly related to decision making, such a result verifies that the aggregated representations exploit also reward information. To sum up, the visualization results show that MASIA can extract valuable global information with these representations.

### 4.3  Study about Individual Information Extraction

To demonstrate the effectiveness of our information extraction module, we analyze the weights computed by the focusing network. Specifically, we select the TJ (medium) task for evaluation and compare the weights produced by two different agents. In this environment, agents could dynamically enter or leave the plane, making the agents staying in the environment flexible through time. It can be observed that agent 3 and agent 6 put focus on similar areas of the aggregated representation. Especially after timestep 15, when agents 3 and 6 are in similar situations and distant from the intersection, their focuses are nearly the same. This verifies that the global state information has been successfully extracted to individual agents. Also, on the top of the figure, we draw a figure to measure the relationship between the cosine similarity of weight vectors of different agents against timesteps. It reveals that the similarity of their focus rises after these two agents begin to proceed in the same line (indicated by the render images posted on the lower parts of Fig. 5 (b)). This also conforms to the intuition that similar messages should be extracted for similar observations.

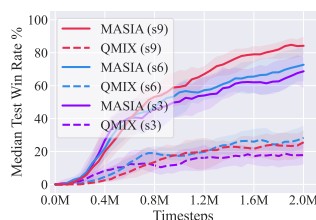

Figure 6: Performance comparison with varying sights, where s$n$ means the sight range is $n$.

### 4.4  Generality of Our Method

We aim to verify that the proposed approach is agnostic to various sight ranges and applied value-based MARL methods. We first conduct experiments on the map 1o10b_vs_1r to show that MASIA could also generalize well on agents with limited observations. The results in Sec. 4.1 show the performance of MASIA when the agents have a sight range of 9. When we narrow the agents' sight ranges, as shown in Fig. 6, by receiving and aggregating messages from teammates, the performance of MASIA does not suffer from a significant drop. Our information aggregation and extraction modules prevent the agent from forfeiting knowledge about the state when the sight range is further limited.

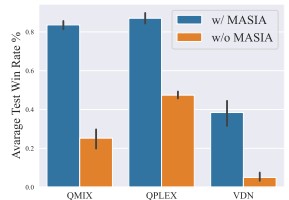

Figure 7: The increase of winning rates brought about by MASIA on map 1o10b_vs_1r.

To show the generality of the MASIA framework, we also carry out experiments to integrate MASIA with current baselines, including VDN, QMIX, and QPLEX. As illustrated in Fig. 7, when integrated with MASIA, the performance of these baselines can be vastly improved on the map 1o10b_vs_1r from SMAC. In this scenario, one overseer is in service to monitor the enemies. Without communication, the other agents have to search the map for the enemies exhaustively. While with reliable communication, they could communicate with each other and the overseer for better coordination. The results demonstrate that MASIA can efficiently aggregate the messages and improve the agents' coordination ability for these value-based MARL methods.

## 5  Related Work

**Multi-agent reinforcement learning (MARL)** has made prominent progress these years. Having emerged under the CTDE paradigm, many methods are designed to relieve the non-stationarity issue, and have made noticeable progress these years. Most of them can be roughly divided into policy-based and value-based methods. Typical policy gradient-methods involves MADDPG [26], COMA [14], MAAC [18], SQDDPG [47], FOP [58], and HAPPO [21] which explore the optimization of multi-agent policy gradient methods, while value-based methods mainly focus on the factorization of the global value function [45, 7]. VDN [42] applies a simple additive factorization to decompose the

joint value function into agent-wise value functions. QMIX [36] structurally enforces the learned joint value function to be monotonic to the agent's utilities, which can represent a more affluent class of value functions. QPLEX [46] further takes a duplex dueling network architecture to factorize the joint value function, achieving a full expressiveness power of IGM [39].

**Multi-agent Communication** Communication plays a promising role in multi-agent coordination under partial observability [11, 60]. Extensive researches have been made on learning communication protocols to improve performance on cooperative tasks [15, 13, 23, 51, 10, 25, 54, 50]. Previous works can be divided into two categories. One focuses on generating a meaningful message for the message senders. The simplest way is to treat the raw local observation, or the local information history as message [13, 40]. VBC [56] and TMC [57] apply techniques, such as variance-based control and temporal smoothing, in the sender end to make the generated messages meaningful and valuable for policy learning. NDQ [48] generates minimized messages for different teammates to learn nearly decomposable value functions, and optimize the message generator based on two different information-theory-based regularizers to achieve expressive communication. On the contrary, other works try to learn efficiently to extract the most useful message on the receiver end, and they design mechanisms to differentiate the importance of messages. I2C [6] and ACML [28] employ gate mechanism to be selective on received messages. There are also works inspired by the broad application of the attention mechanism [2, 5]. TarMAC [4] achieves targeted communication via a simple signature-based soft-attention mechanism, where the sender broadcasts a key encoding the properties of the agents, then the receiver attends to all received messages for a weighted sum of messages for decision marking. SARNet [35] and MAGIC [31] further remove the signature in TarMAC and leverage attention-based networks to learn efficient and interpretable relations between entities, decide when and with whom to communicate.

To the best of our knowledge, none of the existing MARL communication methods explicitly consider how the multiple received messages can be optimized for efficient policy learning. Agents may be confused by redundant information from teammates, and simply augmenting the local policy with the raw message may burden the learning. Our proposed method applies a message aggregation module to learn a compact information representation and extracts the most relevant part for decision-making.

## 6    Conclusion and Future Work

In this paper, we investigate the information representation for multi-agent communication. Previous works either focus on generating meaningful message or designing a mechanism to select the most relevant message in a raw way, ignoring the aggregation of the message, resulting in low sample efficiency in complex scenarios. Our approach improves communication efficiency by learning a compact information representation to ground the true state and optimizing it in a self-supervised way. Also, we apply a focusing network to extract the most relevant part for decision-making. We conduct sufficient experiments in various benchmarks to verify the efficiency of the proposed methods, and more visualization results further reveal why our approach works. For future work, more results on image input and solving the scalability issue when facing environments with hundreds or thousands of agents by techniques like agent grouping would be of great interest.

## Acknowledgements

This work is supported by the National Key Research and Development Program of China (2020AAA0107200), the National Science Foundation of China (61921006, 61876119, 62276126), the Natural Science Foundation of Jiangsu (BK20221442), and the program B for Outstanding PhD candidate of Nanjing University. We thank the anonymous reviewers for their useful suggestions.

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
