# OpenReview forum: "Efficient Multi-agent Communication via Self-supervised Information Aggregation"
_NeurIPS.cc/2022/Conference — NeurIPS 2022 Accept_

### Official Review · Reviewer_Edv9 · 2022-06-14

**Rating:** 7
**Confidence:** 3
**Soundness:** 3 good
**Presentation:** 3 good
**Contribution:** 3 good

**Summary:**

The authors propose MASIA, a method for improving convergence characteristics and communication in cooperative MARL settings.

The key motivation of their work is to produce compact and informative communication by aggregating information from different agents into a single representation, which individual agents can get relevant information from. To incentivize "good" overall representations (depicted as $z_t$ in the paper, the authors introduce a decoding loss (to predict the full state) and a predictive loss (to predict future states via rollout).

In experiments in several standard MARL domains, the authors show that MASIA integrates well with existing MARL training frameworks and typically trains agents faster and more stably than existing methods. Qualitative analysis of emergent representations indicates somewhat interpretable clustering and similarity of communication vectors learned by MASIA agents.

**Questions:**

1) How do the agents behave different during decentralized execution vs. centralized training? I understand the paradigm and have written code using such frameworks. I just want to understand, given the MASIA architectures, what information agents have access to during test time. In particular, I want to understand what feeds into the information aggregation encoder at test time, and if $z_t$ is used at test time as well. I really need to understand this part to assess the technical merits of this work.

2) What are the inputs to the Information Aggregation Encoder generally? The authors use $o_i$ notation, which to me indicates different observations (including over time, with the $t$ superscript). So, for the traffic junction setting, for example, I assume that means the standard observations of things like the location, route id, and surrounding cells? Do the observations include communication, or is communication simply generated afresh as $z_t$ at every timestep based on the history of state observations?

**Limitations:**

The authors could dedicate more time to discussing limitations and societal impacts of their work. I understand that space within the 9-page limit is at a premium; at the very least, including a short discussion in an Appendix would be worthwhile. I simply think that it is an overstatement to say "Besides, we do not see it has obvious negative societal impacts." I can easily list multiple hypothetical scenarios in which, by pooling communication, you allow for adversarial attacks that cover up particular agents' communication, or you regress to mean behaviors instead of allowing for individualized responses. I do not wish to imply that the authors have to address all these concerns, but dismissing them without any discussion in the main paper seems like an overstep.

**Strengths And Weaknesses:**

## Strengths ##

The authors address an important problem: stably training emergent communication. Their proposed solution integrates several distinct ideas (in my interpretation, the autoencoding and rollout objectives represent different concepts) seemingly successfully to train agents that, in experiments, outperform existing methods.

The evaluation is particularly impressive, using a wide variety of settings from existing literature, and reporting median (great job using robust statistics!) values over 5 random seeds.


## Weaknesses ##

Literature review: The authors appear to have missed some relevant related works. The paper could benefit from better framing by citing works on the challenges of emergent communication (e.g., Eccles et al on "Biases for Emergent Communication in Multi-agent Reinforcement Learning"). Most importantly, though, the authors do not appear to know of Lin et al.'s "Learning to Ground Multi-Agent Communication with Autoencoders" from NeurIPS last year. Although Lin's technique is certainly different from this paper's, there are deep underlying similarities, especially regarding the use of an autoencoding loss to promote informative communication. Their method would be an extremely relevant baseline.

Claims about compactness: I am concerned about the authors' claims about compactness. The authors repeatedly note that they seek to induce "compact and sufficient" representations. I understand that an autoencoding loss incentivizes informative, or sufficient, representations by reward information content in representations. However, my understanding of compactness corresponds to notions of low information content, or compression. For example, information-bottleneck methods limit communication information, or using fewer discrete messages creates more "compact" representations. I do not see how the proposed work aligns with this notion. The authors write near line 173 that "By forcing the aggregated representation to be predictive on its future states, the aggregated representation could be more correlated with the information required for its decision-making, which meets the compactness standard." I don't think I agree. Could $z_t$ not simply encode all information about observations and states? If so, that is certainly informative of future states. How does that imply compactness?

Ablation studies: MASIA involves multiple new training objectives over baselines, so I wish I had seen a more complete ablation study to reveal the importance of different components. The studies included in the appendices are a good start: in particular A.3 answers a question I had when reading the paper (although I do actually wonder about the significance of the results, as there seems to be a lot of overlap in error bars and it's only conducted for the very early timesteps). However, there is no such ablation study (as far as I can tell) for the autoencoding loss.  To me, studying the autoencoding loss contribution would be more important than the different pooling mechanisms.

Some lack of clarity: Overall, I felt like I understood most, but not all, of the authors' proposed work. I elaborate on my questions in the next section. I do not mean to imply that the work was entirely unclear, only that there were parts that I did not understand from the main text. Depending on the answers to those questions, I will be better able to assess the technical contributions of the work. Because I am currently unsure about what information agents have access to, I'm not sure of how truly decentralized the agent setup is.

Although certainly not disqualifying on their own, this paper contains some small typos that would have to be fixed for publication. Examples include the caption for Figure 3 "ours" vs. "our", line 221 "injury" should be a verb of some sort.

## Summary ##

Overall, I like the paper and think it could be an important contribution, but I need to make sure I understand aspects of the design better before I recommend acceptance. In other words, I am very willing to "change my mind" because my mind isn't currently made up. I look forward to engaging in followup discussion based on my questions.

---

> ### Author Response · Authors · 2022-08-02
> **Response to reviewer Edv9 (Part 2/2)**
>
> **Q5**: How do the agents behave differently during decentralized execution vs. centralized training? What information do agents have access to during test time?
>
> **A5**: All the further clarification below will be added to our revision. First of all, we would like to emphasize that both in the decentralized execution and centralized training, the agents will first full-communicate their individual observations (determined by the environment observation model). Then each agent feeds the collected observations into the information encoder to obtain an aggregated representation $z_t$ and utilizes the focusing network to extract the most relevant information for its decision-making, which is fed into the subsequent $Q$-network to help estimate the Q-function.
>
> - Thus, coming back to the original question, the differences between the training and testing phases are that we would use state information to help compute the auto-encoder loss and estimate the $Q_\rm{tot}$ (if there is a mixing network in the $Q$-network, e.g., MASIA+QMIX). Besides, multi-step prediction loss is only computed and optimized in the training phase. The state decoder, the latent model, and the possible mixing network are all thrown away during decentralized execution. We remain the Information Aggregation Encoder, the focusing network, and individual $Q$-networks to ensure the decentralized execution process. The representation loss terms that we designed aim to train the encoder network well.
>
> - During test time, each agent can only have access to its individual observation which is defined in the environment (nearby information within sight for the LBF setting).
>
> - What feeds into the information aggregation encoder is all agents' observations which is collected by full-communication. Besides, $z_t$ is used at test time as well as it serves as the aggregated information.
>
> - In the training phase, we only have one Information Aggregation Encoder. After training, the encoder network will be deployed to each agent locally to support decentralized execution.
>
> **Q6**: What are the inputs to the Information Aggregation Encoder generally? Do the observations include communication?
>
> **A6**: As listed in A5, the inputs to the Information Aggregation Encoder are collected observations of all agents. The inputs do not include communication.
>
> **Q7**: The authors could dedicate more time to discussing limitations and societal impacts of their work.
>
> **A7**: We are grateful for this suggestion and have added some discussions about the societal impacts of our work in the revised appendix. As pointed out, each agent needs to collect information from all other agents, and this could open up the possibility of a communication attack and bring negative societal impact. We believe further discussions about adversarial attacks in multi-agent communication reinforcement learning is an interesting direction.
>
> References:
>
> [1] Biases for emergent communication in multi-agent reinforcement learning. NeurIPS, 2019.
>
> [2] Learning to ground multi-agent communication with autoencoders. NeurIPS, 2021.

---

> ### Author Response · Authors · 2022-08-02
> **Response to reviewer Edv9 (Part 1/2)**
>
> Thanks for your detailed comments and interest in our paper.
>
> **Q1**: The authors appear to have missed some relevant related works, including related works on the challenges of emergent communication (e.g., Eccles et al. on “Biases for Emergent Communication in Multi-agent Reinforcement Learning”) and Lin et al.’s “Learning to Ground Multi-Agent Communication with Autoencoders.”
>
> **A1**: Thank you for the reminder about the related works on emergent communication (especially Eccles et al.’s “Biases for Emergent Communication in Multi-agent Reinforcement Learning [1]”) and Lin et al.’s “Learning to Ground Multi-Agent Communication with Autoencoders [2]”. We recognize emergent communication as relevant work and agree that Lin’s technique has similarities with us for using autoencoding loss. Although the two mentioned methods are related to MASIA, they differ from MASIA, [1] is designed to encourage positive signaling and listening between two agents, and [2] aims to employ the learn lingua franca to understand and respond to each other’s utterances. The two mentioned methods are different from our methods, and we will add these related works to our paper and do more discussions to make our literature more sound.
>
> **Q2**: Concerns about compactness-related claims.
>
> **A2**: With the word “compactness,” we are trying to convey that our representations are low in dimension but rich in information. In addition, it also means something slightly different from compression since we abandon the redundant information in the simple concatenation of all agents' observations. For example, the observations of the agents may contain overlapping or repetitive parts. Furthermore, by making the aggregated representation predictive of future states, we make the representations more relevant to agents' decision-making and abandon other useless information. However, rich information related to global state and agents' coordination remains in the representation.
>
> **Q3**: More complete ablation study to reveal the importance of different components, especially for the autoencoding loss that is lacked in paper.
>
> **A3**: Thank you for raising the concerns about the ablation study. To further justify the effectiveness of both the multi-step prediction loss and the autoencoding loss, we additionally conduct ablations for $(1)\lambda_1=0, (2)\lambda_2=0, (3)\lambda_1=\lambda_2=0$ in Hallway and Traffic Junction. For each ablation in each scenario, we run five random seeds and report the mean scalar and standard deviation in the following tables. We provide more detailed descriptions in our revised appendix.
>
> - For Hallway, we adopt both the 4x6x10 and 3x5-4x6x10 tasks. We compare the average number of samples for the ablation to solve the task (the average test win rate of 5 consecutive test timepoints is over 0.95). If any random seed of one ablation fails to solve the task within 2M samples, “failure” is recorded.
> |                                       | $\lambda_1=0,\lambda_2=1$ | $\lambda_1=1,\lambda_2=0$ | $\lambda_1=0,\lambda_2=0$ | $\lambda_1=1,\lambda_2=1$ |
> | ------------------------------------- | ------------------------- | ------------------------- | ------------------------- | ------------------------- |
> | 4x6x10                    | $682.6K\pm241.5K$         | $448.4K\pm88.5K$          | $580.5K\pm152.8K$         | $\textbf{377.0K}\pm12.5K$      |
> | 3x5-4x6x10             | $503.0K\pm183.3K$         | $382.8K\pm46.1K$          | failure                   | $\textbf{265.2K}\pm41.6K$      |
> - For Traffic Junction, additional experiments are conducted on the medium-level task. We compare the ablations' average test win rate performance at 1M, 2M, and 4M timesteps respectively.
> |                           | 1M                     | 2M                    | 4M                     |
> | ------------------------- | ---------------------- | --------------------- | ---------------------- |
> | $\lambda_1=0,\lambda_2=1$ | $0.8125\pm 0.0473$     | $0.8667\pm 0.0514$    | $0.8950\pm0.0292$      |
> | $\lambda_1=1,\lambda_2=0$ | $0.7750\pm0.0652$      | $0.8550\pm0.0534$     | $0.9200\pm 0.0430$     |
> | $\lambda_1=0,\lambda_2=0$ | $0.8000\pm0.0204$      | $0.8583\pm0.0624$     | $0.9167\pm0.0312$      |
> | $\lambda_1=1,\lambda_2=1$ | $\textbf{0.8938}\pm 0.0480$ | $\textbf{0.9000}\pm0.0637$ | $\textbf{0.9225}\pm 0.0375$ |
>
> **Q4**: Some small typos that would have to be fixed for publication
>
> **A4**: Thank you for pointing out the typos in our paper, which helps us improve our work. We modify the “ours” to “our” in Fig. 3 and change the “injury” in Line 221 to “injure.” We have further rechecked the writing of the paper and corrected some additional typos in red.

---

> ### Author Response · Authors · 2022-08-08
> **Dear Reviewer Edv9, have our responses addressed your questions?**
>
> Dear Reviewer Edv9:
>
> We thank you again for your comments and hope our responses could address your questions. As the response system will end soon, please let us know if we missed anything. More questions on our paper are always welcomed. If there are no more questions, we will appreciate it if you can kindly raise the score.
>
> Sincerely yours,
>
> Authors of Paper7996

---

> > ### Comment · Reviewer_Edv9 · 2022-08-08
> > **Thank you**
> >
> > I thank the authors for the careful response.
> >
> > Given the ablation study, I am further inclined to argue for the paper's acceptance and have raised my score from 6 to 7.
> >
> > I think the paper's framing and contribution are strong, but given my (and other reviewers') confusions or questions during the initial review period, I would encourage the authors to pro-actively clarify some common questions in their final version of the paper, should it be accepted.

---

> > > ### Author Response · Authors · 2022-08-08
> > > **Thank you very much**
> > >
> > > We really appreciate the reviewer's comments and suggestions. Your valuable reviews really help us improve the quality of our work. We will take your suggestion and pro-actively clarify these questions in our final version. Thanks very much for your raising the score!

---

### Official Review · Reviewer_t3CW · 2022-07-12

**Rating:** 7
**Confidence:** 4
**Soundness:** 3 good
**Presentation:** 4 excellent
**Contribution:** 3 good

**Summary:**

The authors proposed MASIA which aims to encourage efficient communication among agents within cooperative MARL settings. MASIA uses information aggregator and extractor and learns the message representations that are compact and sufficient for agents’ communication. Internally, self-attention mechanism and predictive representation are used for such representations, where additional model network is used for predictive representation. MASIA is shown to mostly outperform existing MARL baselines on the tasks where agents’ communication is crucial to the overall performance. There are further ablation studies that shows MASIA’s learned features and its compatibility with existing MARL algorithms.

**Questions:**

1. DEC-POMDP is considered to be a theoretical framework for this work, which implies the state s is not directly observable. However, s should be used for the reconstruction loss for information aggregation, which makes me a bit confused. I believe s here seems a set of all agents’ observations under the CTDE, but I hope that the authors clarify this.

2. How each loss affects the performance of MASIA is unclear. From the appendix, I could see that the loss is balanced through the weights $\lambda_1$ and $\lambda_2$. What would be the performance when (1) $\lambda_1=0$ (2) $\lambda_2=0$ (3) $\lambda_1=\lambda_2=0$?


**Limitations:**

There seems no negative societal impact of this research.

**Strengths And Weaknesses:**

Strengths: The authors’ intuitions and approaches are described well. Experiments are done in various perspectives to show the effectiveness of algorithms. Generalizability with existing algorithms is also a strong point.

Weaknesses: The scalability for a large number of agents (e.g., tens or hundreds of agents) is questionable as most of CTDE frameworks has such restriction. I could not find major weaknesses of this work.

---

> ### Author Response · Authors · 2022-08-02
> **Response to reviewer t3CW**
>
> Thanks for your thoughtful comments. Here we offer some explanations to clarify your concerns and we hope they can be helpful.
>
> **Q1**: The scalability for a large number of agents (e.g., tens or hundreds of agents) is questionable as most of CTDE frameworks has such restriction.
>
> **A1**: We strongly agree with your concerns about scalability, a limitation for most CTDE methods. A possible solution to the weakness of scalability is that we can group agents [1] or only consider the neighbors heuristically [2,3] when the agent number is quite large. We will add more discussions to our paper, and we believe it will be a valuable direction for future work.
>
> **Q2**: How is $s$ used in different phases and what does $s$ represent?
>
> **A2**: Thank you for raising doubts about the usage of state $s$ in our paper. Our method follows the CTDE framework [4], and we assume that we can obtain state $s$ in the training phase while state $s$ is not observable when testing. Thus, we assume we can fetch $s$ when computing the reconstruction loss as it happens in training. Depending on the concrete features of the environment, $s$ may be a set of all agents' observations, e.g., in Hallway; $s$ may also be a separate data structure that may contain information out of the set of $o_i$, e.g., in StarCraft2.
>
> **Q3**: What would be the performance when $(1)\lambda_1=0, (2)\lambda_2=0, (3)\lambda_1=\lambda_2=0$?
>
> **A3**: The suggestions for more ablations about the hyperparameters are of great value. We agree that they can further justify the effectiveness of the proposed two representation objectives. We additionally conduct ablations for $(1)\lambda_1=0, (2)\lambda_2=0, (3)\lambda_1=\lambda_2=0$ in the tasks of Hallway and Traffic Junction. These ablation experiments are added to the revised appendix, and a more detailed description can be found there.
>
> - For Hallway, we conduct different ablations in the 4x6x10 and 3x5\-4x6x10 tasks, which are the same as the settings in the original paper. We run five random seeds for each ablation and compare the average number of samples needed to solve the task (the average test win rate of 5 consecutive test timepoints is over 0.95). If any random seed of one ablation fails to solve the task within 2M samples, we record “failure” in the table.
> |                                       | $\lambda_1=0,\lambda_2=1$ | $\lambda_1=1,\lambda_2=0$ | $\lambda_1=0,\lambda_2=0$ | $\lambda_1=1,\lambda_2=1$ |
> | ------------------------------------- | ------------------------- | ------------------------- | ------------------------- | ------------------------- |
> | 4x6x10                    | $682.6K\pm241.5K$         | $448.4K\pm88.5K$          | $580.5K\pm152.8K$         | $\textbf{377.0K}\pm12.5K$      |
> | 3x5-4x6x10             | $503.0K\pm183.3K$         | $382.8K\pm46.1K$          | failure                   | $\textbf{265.2K}\pm41.6K$      |
> - For Traffic Junction, we also perform 5 random seeds for each ablation. We compare the average test win rate performance of each ablation at 1M, 2M, and 4M timesteps for the medium-level scenario.
> |                           | 1M                     | 2M                    | 4M                     |
> | ------------------------- | ---------------------- | --------------------- | ---------------------- |
> | $\lambda_1=0,\lambda_2=1$ | $0.8125\pm 0.0473$     | $0.8667\pm 0.0514$    | $0.8950\pm0.0292$      |
> | $\lambda_1=1,\lambda_2=0$ | $0.7750\pm0.0652$      | $0.8550\pm0.0534$     | $0.9200\pm 0.0430$     |
> | $\lambda_1=0,\lambda_2=0$ | $0.8000\pm0.0204$      | $0.8583\pm0.0624$     | $0.9167\pm0.0312$      |
> | $\lambda_1=1,\lambda_2=1$ | $\textbf{0.8938}\pm 0.0480$ | $\textbf{0.9000}\pm0.0637$ | $\textbf{0.9225}\pm 0.0375$ |
>
> References:
>
> [1] VAST: Value Function Factorization with Variable Agent Sub-Teams. NeurIPS, 2021.
>
> [2] Mean Field Multi-Agent Reinforcement Learning. ICML, 2018.
>
> [3] Concentration Network for Reinforcement Learning of Large-Scale Multi-Agent Systems. AAAI, 2022.
>
> [4] Contrasting Centralized and Decentralized Critics in Multi-Agent Reinforcement Learning. AAMAS, 2021.

---

> ### Author Response · Authors · 2022-08-08
> **Dear Reviewer t3CW, have our responses addressed your questions?**
>
> Dear Reviewer t3CW:
>
> We thank you again for your comments and hope our responses could address your questions. As the response system will end soon, please let us know if we missed anything. More questions on our paper are always welcomed. If there are no more questions, we will appreciate it if you can kindly raise the score.
>
> Sincerely yours,
>
> Authors of Paper7996

---

> ### Comment · Reviewer_t3CW · 2022-08-08
> **Response to Authors**
>
> I appreciate authors for addressing my concerns and will increase my score 6->7.

---

> > ### Author Response · Authors · 2022-08-09
> > **Thanks a lot**
> >
> > We are very grateful for your thoughtful comments. We believe your valuable suggestions can help improve our work, and thank you very much for raising the score!

---

### Official Review · Reviewer_P72Q · 2022-07-14

**Rating:** 5
**Confidence:** 4
**Soundness:** 2 fair
**Presentation:** 2 fair
**Contribution:** 2 fair

**Summary:**

The paper considers multi-agent RL (MARL), where agents share information at each step using communications.  The authors present a centralized training scheme where all the agents observations (sensing) are collected and processed / aggregated, and a focus of attention mechanism is also learned to weight the shared observations for each agent to collect only what is useful of the global information for each agent.  These can be added on to existing MARL methods (section 4.4).  The paper gives an overview the ideas, and then presents several numerical experiments and comparisons.

**Questions:**

Why not just start with a notion of neighbors, and proceed from there?

The comparisons are very difficult to interpret.  For example, “QMIX performs well in LBF, which is attributed to the fact that agents can observe the grids near them, and the mixing network of QMIX can help improve the coordination ability of the fixed group of agents in the training phase.”  So, QMIX knows the state of the neighbors (and communication with them is unnecessary)?

Perhaps some kind of table could be used to compare the different algorithms, including how they communicate or not, the observation model, and what kind of scenarios they fit (or not).

Please clarify “updating of the focusing network is coupled to the RL objective, making the weights produced by the focusing network could be task-sensitive, which could also facilitate policy learning.”

“other communication methods such as TarMAC, NDQ, and TMC achieve low performance or even fail in this environment.”  Why?  Please be specific here, beyond some intuition.

And please clarify “why FullComm succeeds is that there is hardly any redundancy in agents’ observations in Hallway”.

“TarMAC and NDQ have high variance due to the instability of their messages, while MASIA gains high sample efficiency and can generate steady messages since it aims to reconstruct the state.”  What is message instability?


**Limitations:**

No negative social impact is expected.

**Strengths And Weaknesses:**

The idea of learning a focus of attention in the centralized learning phase is interesting.  Basically, what agents should be communicating. The focus of attention is apparently changing with global state (but not well explained).

The presentation is confusing and hard to read, and the algorithm(s) are never clearly laid out. There is undefined notation, and cases where variables are introduced and (perhaps) defined later.  For example, even in the basic model eqn (1), Q, K, and V aren’t clearly defined.

The presentation is weak on mathematical presentation, and many results are described in an intuitive but often confusing way.  There is no theoretical analysis.

The overall scalability is likely to be poor.  Collecting all observations and then reducing them is not scalable.

It’s not quite clear if this work is intended to be model-based or model-free.  When a map is known, and the agent’s states are known, then for example it is straightforward in the global training to set up a local neighborhood for communications.  The algorithm seems to come back to this in the examples, which is not surprising.  In section 2 it is stated that “agents only use local information o_i as message to share within the team” so apparently this is an underlying assumption (but it isn’t clear if agents who are far away from each other but have useful information will be enabled to communicate).

There are a lot of parameters and hyper-parameters, and their selection in general is not considered.

The examples are useful, but also difficult to interpret without significant effort by the reader to examine all the cited works.  For example, it isn’t clear how each algorithm uses communications (or not), and why these are good for some problems and not others.  The conclusions about (local) communications are not surprising in each scenario given.  It seems that when the previously existing algorithm is well matched to the scenario, then there is little be gained with the proposed method.

---

> ### Author Response · Authors · 2022-08-02
> **Response to reviewer P72Q (Part 2/2)**
>
> **Q7**: It seems that when the previously existing algorithm is well matched to the scenario, then there is little be gained with the proposed method.
>
> **A7**: We claim that our method achieves almost the best performance in each task, despite not with a vast range, which shows our algorithm's high applicability and stability, and we believe it is a good contribution.
>
> **Q8**: Why not just start with a notion of neighbors, and proceed from there?
>
> **A8**: We do not make any assumptions about the agents' grouping or neighborhood. The type of communication is broadcast, and agents would send the same message to every other ally agent. Our method aims to help agents efficiently aggregate and process the messages received from all their teammates and extract essential information to improve decision-making and facilitate cooperation. The actual situation should determine whether information from nearby agents is more critical than that from distant agents. For example, when an array of drones encounters an obstacle, the information from the front drone is more critical to the last drone than that from the drone near the last one.
>
> **Q9**: There are some parts of the main text that need further clarification. Does QMIX know the state of the neighbors (and communication with them is unnecessary)?
>
> **A9**: All the further clarification below will be added in our revision. QMIX is a value-based cooperative multi-agent reinforcement learning algorithm based on centralized training and a decentralized execution paradigm. The point we are trying to convey here is that QMIX achieves implicit communication among a fixed group of agents during the centralized training phase and learns a fixed coordination pattern due to the simplicity of LBF. During the decentralized execution phase, QMIX agents can coordinate based on their local view (if any of their teammates are in the view) since they know exactly how their teammates would behave.
>
> **Q10**: To clarify “updating of the focusing network is coupled to the RL objective, making the weights produced by the focusing network could be task-sensitive, which could also facilitate policy learning.”
>
> **A10**: The task here means the abstract subtask assigned to each agent. Agents need to extract the most relevant information to themselves from the aggregated “global information.” The extraction and implicit subtask assignment are all guided by the RL objective and trained in an end-to-end way.
>
> **Q11**: To specify “other communication methods such as TarMAC, NDQ, and TMC achieve low performance or even fail in this environment.”
>
> **A11**: TarMAC, NDQ, and TMC do not aggregate all the information from all agents and extract the most valuable part for each agent. When encountered with a challenging environment, Hallway, where strong coordination is needed, these methods cannot efficiently extract relevant information through communication, so they perform poorly on this benchmark. Further information can be accessed in the table of answer to **Q6** we added to introduce these baselines.
>
> **Q12**: To clarify “why FullComm succeeds is that there is hardly any redundancy in agents’ observations in Hallway”.
>
> **A12**: Among the works about multi-agent communication, they mainly focus on communication in the form of one-to-one or one-to-many. Little work has been done on full communication since full communication often introduces too much redundant information to a single agent. It could pressure the neural network to extract useful information and cause a significant performance drop. However, full communication works fine in Hallway [3], which can be attributed to the fact that the observation of each agent in Hallway only contains the position where it is currently located and its target. There is hardly any redundancy in agents’ messages so that full communication can perform well on Hallway.
>
> **Q13**: To clarify “TarMAC and NDQ have high variance due to the instability of their messages, while MASIA gains high sample efficiency and can generate steady messages since it aims to reconstruct the state.”
>
> **A13**: Message instability indicates that messages in TarMAC and NDQ lack sound support to make them reflective of some true information about the environment. Message representations in TarMAC and NDQ have high variance since they do not serve a steady purpose, while messages in MASIA are destined to reconstruct the global state, which is consistent for all agents.
>
> References:
>
> [1] Attention is all you need. NeurIPS, 2017.
>
> [2] VAST: Value Function Factorization with Variable Agent Sub-Teams. NeurIPS, 2021.
>
> [3] Efficient Multi-Agent Communication via Shapley Message Value. IJCAI, 2022.

---

> > ### Comment · Reviewer_P72Q · 2022-08-09
> > **Thanks for your reply**
> >
> > I appreciate that the value here is in generality and not necessarily enhanced performance. The other reviewers note the generality and do rate this quality very highly.   However, I still wonder about the approach, which requires global information (broadcast), and clearly will not scale and might actually only be applicable for very small scale cases.
> >
> > Q8/A8:  In many problems the agents will only have local communications, and it is natural to build this into the model.  It seems inefficient to apply a method that requires broadcasting, and then learn that only local communications are needed, for cases where the local communications are known a priori.
> >
> > [Personal note: I apologize for not completing this earlier.  I had a death in the family and this pulled me away.  I appreciate the authors response, and the other reviewers points of view.  I have raised my score for this paper.]

---

> ### Author Response · Authors · 2022-08-02
> **Response to reviewer P72Q (Part 1/2)**
>
> Thanks for your inspiring comments. We offer some clarification to your questions here, and we would appreciate it if you had any further comments.
>
> **Q1**: The presentation is confusing and hard to read, and the algorithm(s) are never clearly laid out.
>
> **A1**: Thank you for pointing out the problem with the presentation. In the example pointed out, $Q, K$, and $V$ respectively denote the concepts of query, key, and value in the attention mechanism as defined in [1], in which we give a brief introduction closely after the equation. We will further check each notation in our paper and try to make it easier to understand. Besides, for the overall flow of the algorithm, we provide a pseudo-code in the appendix, which we hope could be helpful.
>
> **Q2**: The presentation is weak on mathematical presentation, and many results are described in an intuitive but often confusing way. There is no theoretical analysis.
>
> **A2**: Our work focuses on fostering multi-agent communication by helping agents aggregate and process messages received more efficiently, and there is not much mathematical formalization and theoretical analysis in our paper. The empirical results demonstrate the efficacy of our method, and we try to offer an explanation of the results for readers. For those too intuitive descriptions, we offer some clarification (see answers to Q9-13) that we hope can be helpful, and we have polished up the statements in our revised paper. Besides, a more theoretical analysis of our method is an interesting direction for future work.
>
> **Q3**: The overall scalability is likely to be poor.
>
> **A3**: We appreciate your concerns about the limitation of scalability in current MARL methods. According to many scenarios we encountered during actual practices, the number of agents is controllable, and considerations about scalability are not a high priority. For scenarios with a large number of agents, we think grouping agents [2] can help alleviate the problems brought by the number of agents. Scalability is a limitation of our method, and we will do more discussions about it in our revision.
>
> **Q4**: It's not quite clear if this work is intended to be model-based or model-free.
>
> **A4**: To be specific, our method is intended to be model-free. The learned model is only utilized to construct model loss to optimize the aggregated representation and our work doe not fit into the category of model-based reinforcement learning. We guess that you have doubts about this part because we skip out the concrete details about the training flow and display them in the appendix. We have emphasized more about it in our revision to guide the readers through the pseudo-code in the appendix. Thank you for this inquiry.
>
> **Q5**: There are a lot of parameters and hyper-parameters, and their selection in general is not considered.
>
> **A5**: In the experiments, we find that our method is not particularly sensitive to the hyperparameters, and we use the same hyperparameters for all four different environments. Our final selection is listed in Section A.3.1 in the appendix. We hope this can be helpful for you.
>
> **Q6**: It isn’t clear how each algorithm uses communications (or not), and why these are good for some problems and not others. Perhaps some kind of table could be used to compare the different algorithms, including how they communicate or not, the observation model, and what kind of scenarios they fit (or not).
>
> **A6**: We offer a table here to compare different algorithms, which is also added to the appendix in our revision. We really appreciate this constructive suggestion.
>
> | **Name**    | **Type of communication** | **Where to process the information (the Sender/Receiver)** | **Matched scenarios**                 |
> |:-------------:|:---------------------------:|:------------------------------------------------------------:|:---------------------------------------:|
> | MASIA(ours) | Full                      | Receiver                                                   | No Restrictions                       |
> | Full-Comm   | Full                      | No                                                         | Without redundant information         |
> | NDQ         | Full                      | Sender                                                     | Value function is nearly decomposable |
> | TarMAC+QMIX | Full                      | Receiver                                                   | Message with relative importance      |
> | TMC         | Time-Partial              | Sender & Receiver                             | Message with transmission loss |
> | QMIX        | No                        | No                                                         | Full observation or easy coordination |

---

> > ### Comment · Reviewer_P72Q · 2022-08-09
> > **Thanks for your reply**
> >
> > I appreciate the authors response.
> >
> > You have pointed out where the notation definitions for equations (1) and (2) come from, but unfortunately these links still aren't in the paper.  It is important to note how the method scales to guide the reader, because scaling with the number of agents is of considerable concern in many applications, as is the need to control the communications and broadcast may be a luxury that is not easily available (and also scales poorly in many practical cases).

---

> ### Author Response · Authors · 2022-08-08
> **Dear Reviewer P72Q, have our responses addressed your questions?**
>
> Dear Reviewer P72Q:
>
> We thank you again for your comments and hope our responses could address your questions. As the response system will end soon, please let us know if we missed anything. More questions on our paper are always welcomed. If there are no more questions, we will appreciate it if you can kindly raise the score.
>
> Sincerely yours,
>
> Authors of Paper7996

---

### Author Response · Authors · 2022-08-02
**Summary of the Revision**

Thank all of you for your thoughtful and inspiring comments. Here is a summary of major updates made to the revision:

1. More detailed descriptions of the training and testing process of our method are added in Appendix A.6.

2. A comparison between different baselines is added in Appendix A.2.

3. More ablations studies about the proposed two representation objectives are conducted in Appendix A.5.

4. Some guidance for readers to Appendix A.6 is added in Line 109-111 to avoid the reader's confusion about the overall flow of our method.

5. Some discussions about the possible societal impact are added in Appendix A.7.

6. Some typos have been corrected.

Note the revision made are marked as “red” in the revised paper.

We hope that our response and revision address your concerns and questions. We are happy to provide further clarification if you have any additional concerns or comments.

---

### Meta-Review · Area_Chair_Nt6U · 2022-08-26

**Recommendation:** Accept
**Confidence:** Less certain

**Metareview:**

The reviewers agree that the main strengths are the generality of the approach, as well as the experimental results (especially after the rebuttal which answered most questions). The overall approach of choosing the attention focus is also well-motivated by specific experiments and makes sense.

A weakness of the paper that has been discussed is the assumption of full broadcast despite considering a decentralized execution scenario, even though in typical applications there are constraints on communication. The high computation cost of the method has similarly been mentioned (and acknowledged by the authors). While these are limitations of the paper, the discussion still ended up in favor of acceptance for the reasons above.



**Award:**

No

---

### Decision · Program_Chairs · 2022-09-14

Accept